# Real-World Incidence of Febrile Neutropenia among Patients Treated with Single-Agent Amrubicin: Necessity of the Primary Prophylactic Administration of Granulocyte Colony-Stimulating Factor

**DOI:** 10.3390/jcm10184221

**Published:** 2021-09-17

**Authors:** Yosuke Dotsu, Hiroyuki Yamaguchi, Minoru Fukuda, Takayuki Suyama, Noritaka Honda, Yasuhiro Umeyama, Hirokazu Taniguchi, Hiroshi Gyotoku, Shinnosuke Takemoto, Ryuta Tagawa, Ryosuke Ogata, Hiromi Tomono, Midori Shimada, Hiroaki Senju, Katsumi Nakatomi, Seiji Nagashima, Hiroshi Soda, Hiroaki Ikeda, Kazuto Ashizawa, Hiroshi Mukae

**Affiliations:** 1Department of Respiratory Medicine, Graduate School of Biomedical Sciences, Nagasaki University, Nagasaki 852-8501, Japan; bb55318030@ms.nagasaki-u.ac.jp (Y.D.); hyamaguchi@nagasaki-u.ac.jp (H.Y.); t-suyama@nagasaki-u.ac.jp (T.S.); maskofstone.n.h@gmail.com (N.H.); msa412u@gmail.com (Y.U.); gyotokuh@nagasaki-u.ac.jp (H.G.); shinnosuke-takemoto@nagasaki-u.ac.jp (S.T.); hmukae@nagasaki-u.ac.jp (H.M.); 2Clinical Oncology Center, Nagasaki University Hospital, Nagasaki 852-8501, Japan; ashi@nagasaki-u.ac.jp; 3Molecular Pharmacology Program and Department of Medicine, Memorial Sloan Kettering Cancer Center, New York, NY 10065, USA; hirokazu_pc@hotmail.co.jp; 4Department of Respiratory Medicine, National Hospital Organization Nagasaki Medical Center, Ohmura 856-8562, Japan; ryuta.tgw@gmail.com (R.T.); tonomohiro@gmail.com (H.T.); nagashima.seiji.kg@mail.hosp.go.jp (S.N.); 5Department of Respiratory Medicine, Sasebo City General Hospital, Sasebo 857-8511, Japan; suke.820@gmail.com (R.O.); dear_tm_1016_612@yahoo.co.jp (M.S.); hsenju@hospital.sasebo.nagasaki.jp (H.S.); h-souda@hospital.sasebo.nagasaki.jp (H.S.); 6Department of Respiratory Medicine, National Hospital Organization Ureshino Medical Center, Ureshino 843-0393, Japan; nktommy-ngs@umin.ac.jp; 7Department of Oncology, Graduate School of Biomedical Sciences, Nagasaki University, Nagasaki 852-8523, Japan; hikeda@nagasaki-u.ac.jp; 8Unit of Translational Medicine, Department of Clinical Oncology, Graduate School of Biomedical Sciences, Nagasaki University, Nagasaki 852-8501, Japan

**Keywords:** amrubicin, chemotherapy, granulocyte colony-stimulating factor, febrile neutropenia, lung cancer

## Abstract

Background: Single-agent amrubicin chemotherapy is a key regimen, especially for small cell lung cancer (SCLC); however, it can cause severe myelosuppression. Purpose: The purpose of this study was to determine the real-world incidence of febrile neutropenia (FN) among patients treated with single-agent amrubicin chemotherapy for thoracic malignancies. Patients and methods: The medical records of consecutive patients with thoracic malignancies, including SCLC and non-small cell lung cancer (NSCLC), who were treated with single-agent amrubicin chemotherapy in cycle 1 between January 2010 and March 2020, were retrospectively analyzed. Results: One hundred and fifty-six patients from four institutions were enrolled. Their characteristics were as follows: median age (range): 68 (32–86); male/female: 126/30; performance status (0/1/2): 9/108/39; SCLC/NSCLC/others: 111/30/15; and prior treatment (0/1/2/3-): 1/96/31/28. One hundred and thirty-four (86%) and 97 (62%) patients experienced grade 3/4 and grade 4 neutropenia, respectively. One hundred and twelve patients (72%) required therapeutic G-CSF treatment, and 47 (30%) developed FN. Prophylactic PEG-G-CSF was not used in cycle 1 in any case. The median overall survival of the patients with FN was significantly shorter than that of the patients without FN (7.2 vs. 10.0 months, *p* = 0.025). Conclusions: The real-world incidence rate of FN among patients with thoracic malignancies that were treated with single-agent amrubicin chemotherapy was 30%. It is suggested that prophylactic G-CSF should be administered during the practical use of single-agent amrubicin chemotherapy for patients who have already received chemotherapy.

## 1. Introduction

Febrile neutropenia (FN) is a major threat to patients that are treated with chemotherapy, as it can result in subsequent hospital admissions, life-threatening infections, treatment delays, and chemotherapy dose reduction. To prevent chemotherapy-related FN, the primary prophylactic use of granulocyte colony-stimulating factor (G-CSF) has been employed. A large-scale meta-analysis of 61 randomized controlled trials of chemotherapy with or without initial G-CSF support revealed that all-cause mortality was lower among patients who received chemotherapy with primary G-CSF support compared to that of without primary G-CSF [1]. The primary prophylactic administration of G-CSF with pegylated granulocyte colony-stimulating factor (PEG-G-CSF) has been approved for the prevention of FN in clinical practice. However, PEG-G-CSF is not used routinely because it is expensive. According to the guidelines developed by the American Society of Clinical Oncology (ASCO) [2], European Organisation for Research and Treatment of Cancer (EORTC) [3], National Comprehensive Cancer Network (NCCN) [4], and Japanese Society of Medical Oncology (JSMO) [5], the prophylactic administration of PEG-G-CSF is recommended during regimens in which the risk of FN is ≥20%.

Amrubicin is a completely synthetic anthracycline derivative, which is characterized by a nine-amino group and a simple sugar moiety. The chemical structure and acute toxicity of amrubicin are similar to those of doxorubicin [6,7]; however, it causes almost no cardiotoxicity [8,9]. Single-agent amrubicin chemotherapy is used to treat small cell lung cancer (SCLC) and non-small cell lung cancer (NSCLC), and it is especially important as the standard second-line regimen for SCLC [10,11]. As the incidence rate of FN in key clinical trials of single-agent amrubicin chemotherapy was lower than 20%, i.e., 14% and 10%, respectively [10,11], the prophylactic administration of G-CSF is not recommended during single-agent amrubicin therapy. However, single-agent amrubicin can cause severe hematological toxicities, such as grade 4 neutropenia and FN, and is associated with a poor prognosis in the clinical setting. In addition, the incidence rates of FN during single-agent amrubicin chemotherapy varied from 0% to 33% in previous studies [10,11,12,13,14,15,16,17,18,19,20,21,22,23,24], and the necessity of the prophylactic use of G-CSF in patients that are treated with single-agent amrubicin in real-world settings remains unclear. 

Based on these results, we conducted a retrospective multi-institutional study, involving patients with thoracic malignancies that were treated with single-agent amrubicin chemotherapy. The primary objective of the study was to determine the real-world incidence rate of FN in this population.

## 2. Patients and Methods

### 2.1. Study Design

This retrospective study was performed at four institutions (Nagasaki University Hospital, Sasebo City General Hospital, National Hospital Organization Nagasaki, and Ureshino Medical Center). The study protocol was reviewed and approved by the ethics committee of each institution. Whenever possible, the fact that the study was being conducted and the purpose of the study were disclosed to the subjects, and they were provided with an opportunity to refuse to participate. This was an independent collaborative (unsponsored) group study. 

### 2.2. Patients and Treatment

The cases of consecutive patients with thoracic malignancies who were treated with single-agent amrubicin between January 2010 and March 2020 were retrospectively analyzed. Medical information regarding the following factors were collected: age; sex; diagnosis; clinical stage; the patients’ history of chemotherapy; the treatment line; the dose of amrubicin (mg/m^2^); pretreatment renal function; the duration of amrubicin therapy; bone marrow toxicities, including FN, leukopenia, neutropenia, thrombocytopenia, and anemia; urinalysis; the results of biochemical tests of renal and hepatic function and electrolyte levels; the use statuses of G-CSF and antibacterial drugs; progression-free survival (PFS); and overall survival (OS). Filgrastim, lenograstim, and nartograstim were used as therapeutic G-CSF drugs. FN was defined as being present in cases in which the patient experienced a single febrile episode involving fever ≥38.0 °C and had an absolute neutrophil count (ANC) of ≤500 cells/mm^3^ (or ≤1000 cells/mm^3^ with an expected decrease ≤ 500 cells/mm^3^). Neutrophil counts were checked each time to determine if a patient had a fever, and clinically expected FNs with only episodes of fever were excluded. Amrubicin was dissolved in 20 mL of normal saline and administered intravenously as a 5 min infusion at a dose of 25–45 mg/mm^2^ on days 1 to 3 every 3–4 weeks. Hematological toxicities were evaluated according to the Common Terminology Criteria for Adverse Events, version 4.03. The data were once collected by the researchers belonging to each hospital, then collected by the central data center, and quality checked, and inquiries about missing items and suspicious sections were addressed. In some cases, the patient was transferred to another hospital during treatment, but we contacted the transferee and collected data. All members were selected as experts who can handle the data properly. 

### 2.3. Statistical Analysis

The primary endpoint was the incidence rate of FN. The secondary endpoints included the duration of hospitalization, whether chemotherapy dose reduction or a treatment delay was required due to hematological adverse events, PFS, and OS. All statistical analyses were performed using IBM SPSS Statistics Advanced, version 27, Japan. Two-sided *p*-values of <0.05 were considered statistically significant. The Kaplan–Meier method was used for the survival analyses of PFS and OS. Welch t and log-rank tests were used for the duration of hospitalization period and survival, respectively. Univariate and multivariate Cox proportional hazards analyses were used for potential risk factors. Progression-free survival was measured from the day chemotherapy commenced until the day the attending physician determined the progression of disease. Overall survival was measured from the day chemotherapy commenced until death by any cause.

## 3. Results

One hundred and fifty-six patients who received single-agent amrubicin chemotherapy were enrolled in this study. The patients’ baseline characteristics are shown in Table 1. All of the patients were included in the evaluations of toxicity and survival. The prophylactic administration of PEG-G-CSF was not performed in any case. All of the patients except one received amrubicin as a second-line or later treatment, and 111 (71%) of the patients had SCLC. Thirty patients (19%) were administered single-agent amrubicin at a full dose of 45 mg/m^2^, and the others (81%) were given a lower dose from cycle 1 onwards. The most commonly used dose was 35 mg/m^2^ amrubicin.

### 3.1. Toxicity

The hematological toxicities that arose during the treatment are listed in Table 2. The most common grade 3 or worse hematological toxicities were neutropenia and leukocytopenia. One hundred and thirty-nine (89%) patients experienced grade 3 or worse hematological toxicities, and 97 (62%) patients experienced grade 4 toxicities. Forty-seven patients (30%) developed FN (Figure 1A). No treatment-related deaths occurred in this study. 

### 3.2. G-CSF Treatment

Therapeutic G-CSF was administered to 112 (72%) patients due to FN (47 patients, 30%) or severe neutropenia (65 patients, 42%) (Appendix A). Prophylactic PEG-G-CSF was not used in cycle 1 in any case. One patient received prophylactic PEG-G-CSF from cycle 2 onwards because he developed severe neutropenia in cycle 1. He was able to continue receiving amrubicin chemotherapy.

### 3.3. Outcomes of FN and Severe Neutropenia

As described previously, 47 patients developed FN due to amrubicin chemotherapy. All of them required antibiotic treatment (Appendix A). In addition, 18 patients experienced treatment delays due to FN, and 15 patients required dose reductions from cycle 2 onwards. Fourteen patients had to be switched to the best supportive care due to reductions in their performance statuses. Data regarding the duration of hospitalization are shown in Figure 1B–D. The duration of hospitalization was significantly longer among the patients who developed FN than among those without FN (29.3 vs. 19.2 days, respectively, *p* < 0.001). The duration of hospitalization was also significantly longer among the patients who experienced severe neutropenia or were treated with therapeutic G-CSF than the patients who did not experience severe neutropenia or were not treated with therapeutic G-CSF. 

### 3.4. Survival Outcomes

The PFS and OS of the patients who received single-agent amrubicin are shown in Figure 2A–H, respectively. The median PFS times of the patients in the FN and non-FN groups were 1.9 (95% confidence interval (CI): 1.3–2.5) and 3.5 (95% CI: 2.4–4.6) months, respectively, and the median PFS time was significantly shorter in the FN group (*p* = 0.003). The median OS times of the patients in the FN and non-FN groups were 7.2 (95% CI: 3.7–10.7) and 10.0 (95% CI: 8.3–11.7) months, respectively, and the median OS time was significantly shorter in the FN group (*p* = 0.025). Neither experiencing grade 4 neutropenia nor being treated with therapeutic G-CSF had a significant impact on PFS or OS. The PFS and OS of 111 patients with SCLC who received single-agent amrubicin are shown in Figure 2I,J, respectively. The median PFS times of the patients who received amrubicin in the FN and non-FN groups were 2.1 and 3.8 months, respectively, and the median PFS time was significantly shorter in the FN group (*p* = 0.021). The median OS times of the patients in the FN and non-FN groups were 7.2 (95% CI: 3.7–10.7) and 10.2 (95% CI: 8.3–11.7) months, respectively, and the median OS time tended to be shorter in the FN group but not significantly different (*p* = 0.072). 

### 3.5. Risk Factors

The potential risk factors of the patients who received single-agent amrubicin for OS were analyzed by univariate Cox proportional hazards and are shown in Table 3. ECOG PS ≥ 2 before treatment and an episode of FN were risk factors for significantly shorter OS in univariate analysis. These two factors and amrubicin dose were analyzed by multivariate Cox proportional hazards and are shown in Table 4. ECOG PS ≥ 2 was a risk factor, but an episode of FN was not in multivariate analysis. A reduction in amrubicin dose of 5 mg/m^2^ before treatment was a risk factor in multivariate analysis. Neither age ≥ 65 nor age ≥ 75 was a risk factor in univariate analysis (Table 3). The potential risk factors of the patients who received single-agent amrubicin for FN were analyzed by Cox proportional hazards and are shown in Table 5 and Table 6. ECOG PS ≥ 2, neutrophils < 2000/µL, and age ≥ 75 before treatment were risk factors that cause FN at a significantly higher frequency in univariate analysis. ECOG PS ≥ 2, and neutrophils < 2000/µL before treatment were risk factors that cause FN at a significantly higher frequency in multivariate analysis. 

## 4. Discussion

In the present study, the real-world incidence rate of FN among patients that were treated with single-agent amrubicin chemotherapy was found to be 30%, which is higher than the 20% cut-off level for the prophylactic use of PEG-G-CSF recommended in the relevant guidelines. Therefore, it is recommended that prophylactic PEG-G-CSF should be administered during single-agent amrubicin chemotherapy for patients with thoracic malignancies. In addition, the patients who developed FN exhibited significantly shorter PFS in SCLC.

Regarding FN, the incidence rates of FN in 17 prospective clinical trials of single-agent amrubicin therapy, involving 1151 patients with lung cancer, are shown in Table 7 [10,11,12,13,14,15,16,17,18,19,20,21,22,23,24,25]. The total FN incidence rate was 12% (n = 1151, range: 0–33%). The studies conducted in Japan and other countries reported FN incidence rates of 14% (n = 625) and 10% (n = 526), respectively. The FN incidence rates in Japan tended to be higher than those seen in other countries, but the incidence rate of FN was below 20% in both types of studies. The incidence rates of FN among patients with SCLC and NSCLC were 11% (n = 780) and 14% (n = 338), respectively. The FN incidence rates in SCLC tended to be lower than those seen in NSCLC because the former included studies in other countries and first-line treatment. Most guidelines, including the ASCO, EORTC, NCCN, and JSMO guidelines, suggest that the prophylactic administration of G-CSF for regimens that carry a high risk of FN (≥20%) can improve OS, but this is not the case for regimens that carry an intermediate risk of FN (10–20%) [2,3,4,5]. Therefore, the prophylactic administration of G-CSF is not currently recommended during single-agent amrubicin chemotherapy. However, we demonstrated that the real-world FN rate was higher than 20%, the patients with FN developed shorter OS, and an episode of FN tends to be a risk factor for shorter OS in the present study. Moreover, the potential risk factors for FN were found to be PS ≥ 2 and neutrophils < 2000/µL via the multivariate Cox proportional hazards analysis. Age ≥ 75 was found to be a potential risk factor for FN via the univariate analysis. The most probable reason for the high frequency of FN in the current study than that previously reported in Table 7 would be whether these were clinical trials or a real-world study. Patients entering clinical trials may have a PS of 0–1 and age restrictions, and are considered to be in good condition. In the actual clinical setting, elderly patients with a PS of 2 and complications present more frequently. We know that there are institutions that provide lung cancer chemotherapy under the policy of not using therapeutic G-CSF. However, we use therapeutic G-CSF when grade 4 neutropenia or FN occurs or is expected to occur, such as on weekends in the current study, so the cause of the high frequency of FN is not due to not using therapeutic G-CSF. 

Dosing of amrubicin shows variability in our study; while the full dose of 45 mg/m^2^ was used for 19% of patients, the most commonly used dose was 35 mg/m^2^ because each attending physician adjusts the dose according to the patient’s condition. This is lower than previous reports of 40 mg/m^2^. Real-world setting studies have shown that doses will be reduced due to an increased proportion of elderly patients (median age 68 years in current study), poor PS (include PS 2 of 25% in current study), and complications compared to previously reported clinical trials. This is considered to be one of the reasons that the FN incidence rate was higher than that of studies in the same country. 

Regarding the administration of prophylactic G-CSF in the present real-world study, none of the 156 patients were given prophylactic PEG-G-CSF in the first amrubicin cycle. Even when all cycles are considered, only one patient received prophylactic PEG-G-CSF during single-agent amrubicin treatment (due to severe neutropenia in the previous cycle). In Japan, the hospitalization costs paid to each institution are fixed according to the type of anticancer drug regimen that the patient receives; thus, their income is not affected by whether PEG-G-CSF, which costs JPY 108,635, is administered. This economic factor and the recommendations outlined in the treatment guidelines have contributed to the fact that PEG-G-CSF is not widely administered during single-agent amrubicin chemotherapy in clinical practice. Nevertheless, as revealed in the present study, the prognosis of the patients who developed FN was significantly shortened to a median of 1.9 months for PFS and a median of 7.2 months for OS (compared with 3.5 and 10 months for the patients without FN, respectively), and a randomized phase III study showed that prophylactic G-CSF was effective in reducing the risk of FN and infections in SCLC patients, despite the addition of prophylactic antibiotics [26]; we should consider the administration of prophylactic PEG-G-CSF. Chemotherapy for thoracic malignancies is increasingly being administered in an outpatient setting, and the administration of primary prophylactic G-CSF will enable safer outpatient chemotherapy [27]. On the other hand, it is also important to consider changes in regimen, drug dose, and drug schedule, instead of primary prophylactic administration of G-CSF if the purpose of chemotherapy is symptom relief. 

Regarding the myelosuppression encountered in the present study, 86% of the patients that were treated with single-agent amrubicin developed grade 3 or worse neutropenia, and 62% experienced grade 4 neutropenia. Among the patients treated with single-agent amrubicin in previous studies, 39–94% developed grade 3 or worse neutropenia, and 18–79% experienced grade 4 neutropenia [10,11,12,13,14,15,16,17,18,19], and efforts are being made to determine appropriate amrubicin doses based on its bone marrow toxicity [28,29]. In studies in which amrubicin was used to treat patients with diseases other than lung cancer, grade 3 or worse neutropenia was reported to occur in 4% of the patients that were treated for thymic tumors and 60% of those that were treated for malignant pleural mesothelioma [30,31]. It might be difficult to complete amrubicin treatment without any problems, as clinicians will want to maintain the standard dose intensity to achieve a strong response. In the current study, 30% of patients developed FN, resulting in longer periods of hospitalization and therapeutic G-CSF and antibiotics being required for longer periods. This suggests that treatment without prophylactic PEG-G-CSF might end up being more expensive, despite the cost of PEG-G-CSF itself being avoided. Furthermore, the development of FN due to amrubicin therapy can adversely affect patients’ performance statuses and quality of life. Thirteen percent of the patients in the present study were unable to receive continuous chemotherapy due to FN. This demonstrated that FN can have an impact on survival in some cases. Consequently, we should consider the prophylactic administration of PEG-G-CSF during single-agent amrubicin chemotherapy for thoracic malignancies. 

Regarding the risk factors in the present study, PS ≥ 2 was a risk factor for shorter OS. It is interesting to note that single-agent amrubicin treatment in elderly patients was not a risk factor in the present study, whether they were 65 years or older or 75 years or older. Although chemotherapy in elderly patients is known to have different results than in younger patients [32,33,34], it is presumed that chemotherapy can safely be administered to even elderly patients with good PS and an appropriate dose setting. An amrubicin dose of 45 mg/m^2^ was a factor for longer OS, and decrease in dose before treatment was a risk factor. This may be attributed to not only the higher efficacity of the high dose of amrubicin, but also to the attending physician’s belief that the patients had sufficient organ function, were young enough, and had a sufficiently small number of complications to be able to withstand full-dose amrubicin treatment. Patients whose dose of amrubicin is reduced may have same cause, such as poor general condition, which the attending physician considers necessary to reduce the dose. Thus, amrubicin dose is considered to be a confounding factor.

This study had several limitations. First, it was a small-scale investigation conducted at four institutions, and therefore, it was not possible to draw definitive conclusions. However, it can provide some hypotheses for future research. An external validation study involving a larger number of patients might be needed to confirm our findings. Second, this study was only conducted in one country (Japan). The FN rates in Japan tend to be higher than those observed elsewhere, as shown in Table 7, and it is hoped that studies will be conducted in real-world settings in other regions as well. Third, we did not evaluate the cost-effectiveness of using prophylactic PEG-G-CSF. The primary prophylactic use of PEG-G-CSF is expensive compared to therapeutic G-CSF treatment, and this issue should be considered in future. 

In conclusion, the real-world incidence rate of FN among patients with thoracic malignancies that were treated with single-agent amrubicin chemotherapy was 30%. It is suggested that prophylactic G-CSF should be administered during the practical use of single-agent amrubicin for patients who have already received chemotherapy. 

## Figures and Tables

**Figure 1 jcm-10-04221-f001:**
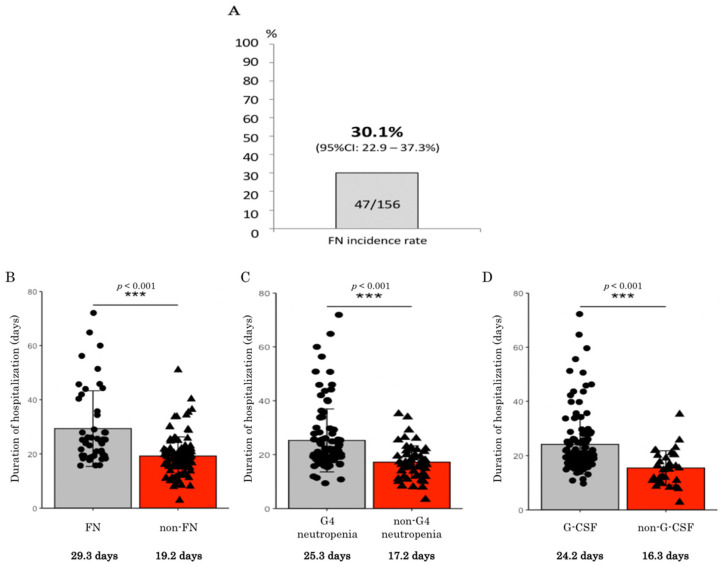
(**A**) Febrile neutropenia (FN) incidence rate in 156 patients who received single-agent amrubicin chemotherapy. (**B**) Duration of hospitalization period among patients that received amrubicin chemotherapy with or without febrile neutropenia (FN). (**C**) Duration of hospitalization period among patients that received amrubicin chemotherapy with or without grade 4 (G4) neutropenia. (**D**) Duration of hospitalization period among patients that received amrubicin chemotherapy with or without G-CSF. ***: statistically significant difference; G-CSF: granulocyte colony-stimulating factor; CI: confidence interval.

**Figure 2 jcm-10-04221-f002:**
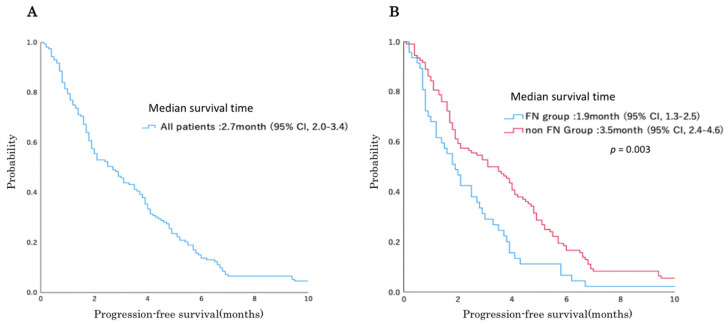
Survival curves using the Kaplan–Meier method for patients treated with amrubicin. (**A**) Progression-free survival curve of all 156 patients. (**B**) Progression-free survival curve of patients with or without FN. (**C**) Progression-free survival curve of patients with or without grade 4 (G4) neutropenia. (**D**) Progression-free survival curve of patients treated with or without G-CSF. (**E**) Overall survival curve of all 156 patients. (**F**) Overall survival curve of patients with or without FN. (**G**) Overall survival curve of patients with or without grade 4 neutropenia. (**H**) Overall survival curve of patients treated with or without G-CSF. (**I**) Progression-free survival (PFS) curve and median survival times of 111 small cell lung cancer patients with or without febrile neutropenia (FN). (**J**) Overall survival (OS) curve of 111 small cell lung cancer patients with or without FN.

**Table 1 jcm-10-04221-t001:** Patient characteristics (*N* = 156).

Characteristics	Number	(%)
Age, years		
Median	68	
Range	32–86	
Sex		
Female	30	(19)
Male	126	(81)
ECOG PS		
0	9	(6)
1	108	(69)
≥2	39	(25)
Histology		
Small cell carcinoma	111	(71)
Adenocarcinoma	20	(13)
Mesothelioma	8	(5)
LCNEC	7	(5)
Squamous cell carcinoma	3	(2)
Others	7	(5)
Stage		
III	2	(1)
IV	140	(90)
Recurrence	14	(9)
No. of prior chemotherapies		
0	1	(1)
1	96	(61)
2	31	(20)
≥3	28	(18)
Neutrophil count (/µL) before AMR therapy		
Median	3860	
Range	1300–13,500	
AMR dose (mg/m^2^)		
25	3	(2)
30	26	(17)
35	70	(45)
40	27	(17)
45	30	(19)

Abbreviations: ECOG PS, Eastern Cooperative Oncology Group performance status; LCNEC, large cell neuroendocrine carcinoma; AMR, amrubicin;

**Table 2 jcm-10-04221-t002:** Hematological toxicities (*N* = 156).

Adverse Events	≥Grade 3	≥Grade 4
Leukopenia	111 (71%)	52 (33%)
Neutropenia	134 (86%)	97 (62%)
Anemia	35 (22%)	2 (1%)
Thrombocytopenia	36 (23%)	11 (7%)
Febrile neutropenia	47 (30%)	-

**Table 3 jcm-10-04221-t003:** Potential risk factors for OS by the univariate Cox proportional hazards analysis (*N* = 156).

Characteristics	HR	95% CI	*p*-Value
Age, years			
<65	1		
≥65	0.815	0.552–1.204	0.305
Age, years			
<75	1		
≥75	0.859	0.532–1.386	0.533
Sex			
Female	1		
Male	1.083	0.680–1.725	0.736
ECOG PS			
0–1	1		
≥2	3.533	2.276–5.485	<0.001
Histology			
Small cell carcinoma	1		
Others	1.140	0.755–1.721	0.533
Stage			
IV	1		
Recurrence	0.935	0.499–1.751	0.834
Prior chemotherapies			
0–1	1		
≥2	1.178	0.810–1.715	0.392
AMR dose (mg/m^2^)			
45	1		
Dose decreased by 5	1.731	0.994–3.017	0.053
FN			
Non-FN	1		
FN	1.587	1.055–2.388	0.027
Grade 4 neutropenia			
Non-G4N	1		
G4N	1.216	0.828–1.784	0.319
GCSF			
Without G-CSF	1		
With G-CSF	1.275	0.816–1.991	0.286

Abbreviations: OS, Overall Survival; ECOG PS, Eastern Cooperative Oncology Group performance status; AMR, amrubicin; FN, febrile neutropenia; G4N, grade 4 neutropenia.

**Table 4 jcm-10-04221-t004:** Potential risk factors for OS by the multivariate Cox proportional hazards analysis (*N* = 156).

Characteristics	HR	95% CI	*p*-Value
ECOG PS			
0–1	1		
≥2	3.933	2.395–6.458	<0.001
AMR dose (mg/m^2^)			
45	1		
Dose decreased by 5	1.816	1.033–3.193	0.038
FN			
Non-FN	1		
FN	1.551	0.969–2.484	0.067

Abbreviations: OS, Overall Survival; HR, Hazard Ratio; ECOG PS, Eastern Cooperative Oncology Group performance status; AMR, amrubicin; FN, febrile neutropenia.

**Table 5 jcm-10-04221-t005:** Potential risk factors for FN by the univariate Cox proportional hazards analysis (*N* = 156).

Characteristics	HR	95% CI	*p*-Value
Age, years			
<65	1		
≥65	0.641	0.298–1.377	0.254
Age, years			
<75	1		
≥75	2.22	1.010–4.883	0.047
Sex			
Female	1		
Male	1.233	0.505–3.011	0.646
ECOG PS			
0–1	1		
≥2	3.02	1.415–6.444	0.004
Histology			
Small cell carcinoma	1		
Others	1.227	0.575–2.620	0.597
Stage			
IV	1		
Recurrence	1.497	0.393–5.706	0.555
Prior chemotherapies			
0–1	1		
≥2	0.667	0.324–1.373	0.271
AMR dose (mg/m^2^)			
45	1		
Dose increased by 5	1.522	0.641–3.615	0.341
Neutrophils before treatment			
≥2000/µL	1		
<2000/µL	3.417	1.189–9.822	0.023
Neutrophils before treatment			
≥2500/µL	1		
<2500/µL	0.545	0.236–1.256	0.154

Abbreviations: OS, Overall Survival; ECOG PS, Eastern Cooperative Oncology Group performance status; AMR, amrubicin; FN, febrile neutropenia.

**Table 6 jcm-10-04221-t006:** Potential risk factors for FN by the multivariate Cox proportional hazards analysis (*N* = 156).

Characteristics	HR	95% CI	*p*-Value
ECOG PS			
0–1	1		
≥2	3.018	1.360–6.694	0.007
Age, years			
<75	1		
≥75	1.713	0.739–3.973	0.21
Neutrophils before treatment			
≥2000 /µL	1		
<2000 /µL	3.866	1.281–11.671	0.016

Abbreviations: FN, febrile neutropenia; ECOG PS, Eastern Cooperative Oncology Group performance status.

**Table 7 jcm-10-04221-t007:** Incidence rate of febrile neutropenia after single-agent amrubicin therapy in previous prospective studies (*N* = 1151) (10–25).

Author	Year	Phase	Disease	Number	FN (%)
Onoda (12)	2006	II	SCLC	60	3 (5)
Yana (13)	2007	II	SCLC	33	0 (0)
Igawa (14)	2007	II	SCLC	27	4 (15)
Inoue (10)	2008	II	SCLC	29	4 (14)
Ettinger (15)	2010	II	SCLC	69	8 (12)
Kaira (16)	2010	II	SCLC	29	1 (3)
Jotte (17)	2011	II	SCLC	49	5 (10)
Pawel (11)	2014	III	SCLC	408	41 (10)
Murakami (18)	2014	II	SCLC	82	22 (27)
Inoue (19)	2015	II	SCLC	27	5 (19)
Igawa (20)	2008	II	NSCLC	39	2 (5)
Kaneda (21)	2010	II	NSCLC	61	18 (30)
Kaira (16)	2010	II	NSCLC	37	0 (0)
Yoshida (22)	2011	II	NSCLC	18	6 (33)
Kitagawa (23)	2012	I	NSCLC	16	1 (6)
Yoshioka (24)	2017	III	NSCLC	98	13 (13)
Saigusa (25)	2019	II	NSCLC	69	9 (13)
Total				1151	142 (12)

Abbreviations: FN, febrile neutropenia; SCLC, small cell lung cancer; NSCLC, non-small cell lung cancer.

## Data Availability

The datasets generated and analyzed during the study are available from the corresponding author on reasonable request.

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
