# Peer review of "Real-World Incidence of Febrile Neutropenia among Patients Treated with Single-Agent Amrubicin: Necessity of the Primary Prophylactic Administration of Granulocyte Colony-Stimulating Factor"

_jcm, 2021, doi:10.3390/jcm10184221_

Round 1
Reviewer 1 Report
Dear Authors.
Thank you for this work - I enjoyed reading it. While to many physicians this seems an uninteresting study, I feel that your work is important.
The use of white cell growth factors to prevent febrile neutropenia (FN) and myelosuppression that causes dose reductions and treatment delays impacts on treatment outcomes for patients. Without data such as yours, we would not know the FN rates of chemotherapy regimens in real world settings.
Pivotal trials of new agents tend to exclude older patients and those with comorbidity or a significant symptom burden that decreases patient performance. Since these are all well-recognised risk factors for FN, we should not be surprised that those trials underestimate toxicity that we will see in routine practice. Without such data as you have presented, we would not have the evidence base to recommend whether patients should receive primary prophylaxis with filgrastim, or keep it in reserve for secondary prophylaxis once FN has occurred or unacceptable myelosuppression has been seen.
May I make just one suggestion for the English in the paper to help readability and precision:
For the sentence = "About dose of amrubicin, while the full dose of 45 mg/m2 was also used at 19%, the most commonly used dose in current study was 35 mg/m2 because each attending physician adjusts the dose according to the patient’s condition". Can I instead suggest "Dosing of amrubicin shows variability in our study; while the full dose of 45 mg/m2 was used for 19% of patients, the most commonly used dose was 35 mg/m2 because each attending physician adjusts the dose according to the patient’s condition".
The only reason that an editor might not accept your paper is that amrubicin (a third-generation synthetic anthracycline) is approved currently in Japan (because of its favourable cardiac risk profile and activity in second and subsequent lines of lung cancer treatment) but is not approved currently in Europe or the USA. I personally think that this is not an issue because the most important point is that real world data is needed for all chemotherapy regimens to assign them to the correct FN risk factor. You make a good point that this may not be the same for all patient groups or for all races or countries. This emphasises the importance of your work. This is the greater message of your paper. Please consider this if you are unlucky enough not to have your paper selected for publication.
Lastly, you comment that the reason why primary prophylaxis is often not prescribed is because of the cost. The recent approval of biosimilar versions of pegfilgrastim in many nations (for example Europe, the USA, but not yet Japan) will reduce costs significantly and should signal that health-economic decisions about primary prophylaxis may need to be reassessed at the biosimilar price point.
Best wishes and kind regards - Your Reviewer
Author Response
Thank you for all the encouragement and helpful comments. We follow your indication, and the sentence of “About dose of amrubicin, while the full dose 45 mg/m2 was also used at 19%, the most commonly used dose in current study was 35 mg/m2 because each attending physician adjusts the dose according to the patient’s condition” was changed to “Dosing of amrubicin shows variability in our study; while the full dose of 45 mg/m2 was used for 19% of patients, the most commonly used dose was 35 mg/m2 because each attending physician adjusts the dose according to the patient’s condition”.
Reviewer 2 Report
The present study has several limitations that cannot be overcome. In fact, there are no guidelines for retrospective studies, the methodology is poor and also the structure is not correct.
Author Response
Thank you for the peer review. Certainly, prospective studies give the most accurate results. On the other hand, the other events are happen in real-world general practice. Since strong myeloid toxicity occurs when this drug is administered at the optimal dose determined in clinical trials, the attending physician should reduce the dose according to each idea. Even so, it is highly toxic, and the result is as shown in this paper. As you pointed out, retrospective studies have various limitations. This paper has been reviewed by several medical experts and statisticians, and has been rewritten many times according to the advice. Both English and small mistakes have been pointed out and improved. Despite the limitations of retrospective research, we are confident that this paper is finally completed form.
Round 2
Reviewer 2 Report
The changes have not been tracked. The checklist for retrospective studies was not added and developed.
Author Response
Semtember 8, 2021
Dear Reviewer,
Thank you for your important comments. We are grateful and revised the paper as below.
Sincerely,
Minoru Fukuda, MD, PhD
Clinical Oncology Center, Nagasaki University Hospital
Response to Reviewer 2:
“The changes have not been tracked. The checklist for retrospective studies was not added and developed.”
→ Thank you for the peer review. We searched for a checklist of retrospective studies and found the paper. Motheral B et al. reported “A checklist for retrospective database studies – report of the ISPOR track force on retrospective databases” in ISPOR 2003, and it is written in the form of 27 questions. We checked as follows. In English language, we asked Medical English Service (med-english.com, Kyoto, Japan) and English proofreading has done.
Data Sources
✔️1. Relevance: Have the data attributes been described in sufficient detail for decision makers to determine whether there was a good rationale for using the data source, the data source’s overall generalizability, and how the findings can be interpreted in the context of their own organization?
→ The study was performed at four institutions (Nagasaki University Hospital, Sasebo City General Hospital, National Hospital Organization Nagasaki, and Ureshino Medical Center) which we belong and have good rationale for using the data source. It is added to the 2.1. Study design.
✔️2. Reliability and validity: Have the reliability and validity of the data been described, including any data quality checks and data cleaning procedures?
→ The data was once collected by the researchers belonging to each hospitals, collected by the central data center, and quality checked, and inquiries about missing items and suspicious parts were improved. It is added to 2.2. Patients and treatment.
✔️3. Linkages: Have the necessary linkages among data sources and/or different care sites been carried out appropriately, taking into account differences in coding and reporting across sources?
→ Data is properly sent from each hospitals to the central data center. In some cases, the patient was transferred to another hospital during treatment, but we contact the transferee and collect data. It is added to 2.2. Patients and treatment.
✔️4. Eligibility: Have the authors described the type of data used to determine member eligibility?
→ All members were selected as experts who can handle the data properly. It is added to 2.2. Patients and treatment.
Methods
Research design
✔️5. Data analysis plan: was a data analysis plan, including study hypotheses, developed a priori?
→ Data analysis plan was properly peer-reviewed by institutional review board of the ethics committee and developed a priori. It is added to 2.1. Study design.
✔️6. Design selection: has the investigator provided a rationale for the particular research design?
→ We selected a retrospective multi-institutional study of 4 institutions and a rationale was described in Introduction.
✔️7. Research design limitations: did the author identify and address potential limitations of that design?
→ We described the research design limitations in Discussion.
✔️8. Treatment effect: for studies that are trying to make inferences about the effects of an intervention, does the study include a comparison group and have the authors described the process for identifying the comparison group and the characteristics of the comparison group as they relate to the intervention group?
→ No comparison group was created in this study. We are using existing prospective studies as historical controls. It is described in the Discussion.
✔️9. Sample selection: have the inclusion and exclusion criteria and the steps used to derive the final sample from the initial population been described?
→ The cases of consecutive patients with thoracic malignancies who were treated with single-agent amrubicin between January 2010 and March 2020 were retrospectively analyzed. It was described in 2.2. Patients and treatment.
✔️10. Eligibility: are subjects eligible for the time period over which measurement is occurring?
→ Patients were eligible for the period. It was described in 2.2. Patients and treatment.
✔️11. Censoring: were inclusion/exclusion or eligibility criteria used to address censoring and was the impact on study findings discussed?
→ Cases that have been treated with amrubicin for even one cycle during the specified period are made eligible. There are no censoring case for eligibility.
✔️12. Operational definitions: are case (subjects) and end point (outcomes) criteria explicitly defined using diagnosis, drug markers, procedure codes, and/or other criteria?
→ It was added that “Progression-free survival was measured from the day chemotherapy commenced until progression the day the attending physician determined the progression disease. Overall survival was measured from the day chemotherapy commenced until death by any cause” to 2.3 statistical analysis.
✔️13. Definition validity: have the authors provided a rationale and/or supporting literature for the definitions and criteria used and were sensitivity analyses performed for definitions or criteria that are controversial, uncertain, or novel?
→ In 2.3. Statistical analysis, we described the definitions.
✔️14. Timing of outcome: is there a clear temporal (sequential) relationship between the exposure and outcome?
→ There is a relationship between the exposure (amrubicin) and outcome (febrile neutropenia rate), and we are careful not to lack data.
✔️15. Event capture: are the data, as collected, able to identify the intervention and outcomes if they actually occurred?
→ Febrile neutropenia (FN) occurred in hospital and patient’s home, and the latter may cause data loss. In most cases, FN will not be overlooked as the patient will go to the outpatient department of hospital. In this study, we are dealing only with events that can be captured on electronic medical records. Neutrophil counts were checked each time to determine if a patient had a fever, and clinically expected FNs with only episodes of fever were excluded. It was described to 2.2. Patient and treatment.
✔️16. Disease history: is there a link between the natural history of the disease being studied and the time period for analysis?
→ This study observed overall and progression-free survival, not only because FN occurred or did not occur, but it is related to the natural history of the disease of lung cancer and other factors. Thus, we analyzed potential risk factors as Tables 3 and 4.
✔️17. Resource valuation: for studies that examine costs, have the authors defined and measured an exhaustive list of resources affected by the intervention given the perspective of the study and have resource prices been adjusted to yield a consistent valuation that reflects the opportunity cost of the resource?
→ This is an important point. However, this time we could not evaluate the cost. It is stated in the limitation paragraph of Discussion.
✔️18. Control variables: if the goal of the study is to examine treatment effects, what methods have been used to control for other variables that may affect the outcome of interest?
→ Although the goal of the study is not to examine treatment effect, we analyzed potential risk factors as Tables 3 and 4 to control for other variables that may affect the outcome of interest.
✔️19. Statistical model: have the authors explained the rationale for the model/statistical method used?
→ Two-sided p values of <0.05 were used considered statistically significant for duration of hospitalization. We used it to detect whether the difference between the null hypothesis and the alternative hypothesis is large or small. Kaplan-Meier method and log-rank tests were used for survival analysis because they are common for survival curve analysis. Cox proportional hazard analysis was used because multivariate analysis of survival analysis. They were described in 2.3. Statistical analysis.
✔️20. Influential cases: have the authors examined the sensitivity of the results to influential cases?
→ In this study for primary endpoint, we had two choices, whether to cause FN or not, and since it is considered that there are no outliers, we did not examine the sensitivity to influential cases. Similarly, the impact of outliers is not considered to be that great in analysis of survival and risk factor. In secondary endpoints of duration of hospitalization, all values are displayed in the figure in consideration of the possibility of outliers.
✔️21. Relevant variables: have the authors identified all variables hypothesized to influence the outcome of interest and included all available variables in their model?
→ We identified all variables hypothesized to influence and included for patient characteristics and results.
✔️22. Testing statistical assumptions: do the authors investigate the validity of the statistical assumptions underlying their analysis?
→ We did not investigate the validity of the statistical assumptions.
✔️23. Multiple tests: if analyses of multiple groups are carried out, are the statistical tests adjusted to reflect this?
→ In order to prevent erroneous results from being derived by multiple tests, we have set up the protocol to the institutional review board, and after approved, we are conducting tests all at once.
✔️24. Model prediction: if the authors utilize multivariate statistical techniques in their analysis, do they discuss how well the model predicts what it is intended to predict?
→ We did not discuss about the model prediction.
Discussion/Conclusions
✔️25. Theoretical basis: have the authors provided a theory for the findings and have they ruled out other plausible alternative explanations for the findings?
→ We provides a theory that higher FN rate and ruled out other explanations for findings in discussion.
✔️26. Practical versus statistical significance: have the statistical findings been interpreted in terms of their clinical or economic relevance?
→ In retrospective database studies, the sample sizes are often extremely large, which can render potentially unmeaningful differences to be statistically significantly different. However, the sample size is not that large in present study, so we do not think this checklist is a concern.
✔️27. Generalizability: have the authors discussed the populations and settings to which the results can be generalized?
→ Of course, the possibility of generalization of present study is high if it is a general clinical practice in the same country, however, the situation may be different in other countries. These are described in limitation of Discussion.